# "I told myself, be bold and go and test": Motivators and barriers to HIV testing among gay, bisexual, and other cis-gender men who have sex with men in Ghana – West Africa

Gamji Rabiu Abu-Ba'are[1,2,3,4]*, Gloria Aidoo-Frimpong[3], Melissa Stockton[5], Edem Yaw Zigah[4,6], Samuel Amuah[7], Prince Amu-Adu[6], Richard Panix Amoh-Otoo[7], Laura Nyblade[8], Kwasi Torpey[9], LaRon E. Nelson[3,10]

1 Behavioral, Sexual and Global Health Lab, School of Nursing, University of Rochester Medical Center, University of Rochester, Rochester, New York, United States of America, 2 Department of Public Health Sciences, University of Rochester Medical Center, University of Rochester, Rochester, New York, United States of America, 3 Yale AIDS Prevention Program (Y-APT), Center for Interdisciplinary Research on AIDS, School of Public Health/Medicine, Yale University, New Haven, Connecticut, United States of America, 4 Behavioral, Sexual and Global Health Lab, West Africa Site, Jama'a Action, West Legon, Accra, Ghana, 5 Department of Epidemiology, Gillings School of Global Public Health, University of North Carolina at Chapel Hill, Chapel Hill, North Carolina, United States of America, 6 Priorities on Rights and Sexual Health, Accra, Ghana, 7 Youth Alliance for Health and Human Rights, Kumasi, Ghana, 8 RTI International, Washington, District of Columbia, United States of America, 9 Department of Population, Family & Reproductive Health, School of Public Health, University of Ghana, Legon-Accra, Ghana, 10 School of Nursing, Yale University, New Haven, Connecticut, United States of America

* GamjiRabiu_Abubaare@URMC.Rochester.edu

**Data Availability Statement:** The qualitative data illustrating the findings of the study are presented

## Abstract

Despite a disproportionately high burden of HIV, GBMSM in Ghana and sub-Saharan Africa often delay testing until the point of illness. However, limited studies examine factors that affect their participation in testing. We used qualitative in-depth interviews (IDIs) and focus group discussions (FGDs) to collect insights into experiences, motivators, and barriers to HIV testing among GBMSM. Two community-based organizations used snowball and convenience sampling to recruit 10 GBMSM for IDIs and 8 to 12 for FGDs. We transcribed, coded, identified, and analyzed the relationship and commonalities between the participants' responses. Under experiences with testing, 1) fear of HIV infection created a stressful HIV testing experience, and 2) a friendly and supportive healthcare environment facilitated a positive experience in healthcare facilities. Motivators or facilitators of testing include 1) the perception or belief that HIV testing is an HIV prevention strategy; 2) encouragement from friends and peers; 3) understanding risk associated with certain sexual behaviors; 4) education or information on HIV; 5) access to free testing and incentives; 6) early symptoms and provider recommendation. Barriers to HIV testing include 1) negative community perceptions of HIV; 2) individual-level low-risk perception or indifference about HIV infection; 3) health system issues; 5) Perceived stigma at healthcare facilities. The findings point to the need to address critical issues around stigma, education, peer support, and healthcare resources through interventions and research to improve HIV testing among GBMSM in the country.

as participants' quotes within the paper. The raw datasets used and/or analyzed during the current study are not publicly available due to our ethical and legal requirements for protecting participant privacy and current ethical institutional approvals but are available from the corresponding author on reasonable request pending ethical approval. The publishing of the raw data set is limited due to the high risk of persecution and severe adverse social consequences related to the socio-political sensitivity of the topic of same-sex behaviors in Ghana.

**Funding:** The funding for this study was awarded to LEN, LN, and KT by the National Institute of Nursing Research, grant number R01NR019009. GRA and GA time working on the study was supported by The Yale Center for Interdisciplinary Research on AIDS (P30 MH06224), and The Yale AIDS Prevention Training program (T32 MH020031). The funders had no role in study design, data collection and analysis, decision to publish, or preparation of the manuscript.

**Competing interests:** The authors have declared that no competing interests exist.

## Introduction

In Ghana, gay, bisexual, and other cisgender men who have sex with men (GBMSM) are at heightened risk for HIV infection [1, 2]. As of 2021, GBMSM had 28 times the risk of acquiring HIV compared to heterosexual men [3]. Also, 17.1% of GBMSM live with HIV compared to 2% nationwide [4]. Ghana and other SSA, with the aid of international health organizations (e.g., USAID, PEPFAR), have developed targeted behavioral surveillance interventions to reach GBMSM and increase testing, use of prevention, and linkage to care among GBMSM [1–5]. However, several factors, such as low HIV knowledge, condomless anal sex, discrimination and criminalization, and social isolation continue to hinder HIV prevention and care among GBMSM [1–10].

At the GBMSM level, limited studies in Ghana show that the majority of GBMSM in the country have a lower understanding of HIV prevention and treatment processes [10–14]. For instance, a study among 137 GBMSM showed low mean scores for knowledge of STI and HIV transmission, prevention, and treatment [14]. As such, there is lower use of prevention tools such as condoms, lubricants, and HIV testing, and in some cases, limited adherence to medication. Also, some GBMSM who have sex with women use more condoms when having vaginal sex and less when having anal sex with men [14]. Most did not use condoms during their first sexual encounters and do not use condoms with current sexual partners [14]. These increased risk behaviors and high HIV prevalence among GBMSM show the need for increased HIV testing [10–19].

In addition to low HIV knowledge and access to HIV prevention and care services, high HIV and sexual stigma and discrimination affect HIV testing among GBMSM [15, 16]. Recognizing the centrality of Stigma in HIV testing practices, this study employs the Health Stigma and Discrimination Framework (HSDF) to unpack GBMSM's experiences across the socio-ecological model [20]. HSDF posits that the stigmatization process unfolds across the socio-ecological spectrum in the context of a health condition, such as HIV care, and ultimately impacts uptake of testing, adherence and treatment, incidence, morbidity, and mortality among affected groups [20]. Consistent with the HSDF, Ghanaians generally have a high misconception of HIV and treatment and view HIV as a death sentence. Hence, many GBMSM have a fear of a positive diagnosis [4, 9–16].

Further, same-sex sexual behaviors can be treated as criminal in Ghana, and a recent national anti-LBGTQ political rhetoric hampers service delivery and engagement in HIV prevention programming [21, 22]. As such, GBMSM face stigma at the community and healthcare facility levels, manifesting through gossip, verbal and physical abuse, blame, shame, and ultimately, poor quality of healthcare services [9, 10, 23]. All these manifestations of stigma occur at higher levels for GBMSM who also have gender non-conforming expressions (for example, men who act in ways that will be considered feminine in the Ghanaian context), not only from the community but healthcare providers and at the interpersonal level between GBMSM peers [9, 10, 23].

Regular HIV testing is vital to both HIV prevention and linkage to care; HIV testing allows GBMSM to be aware of their HIV status, facilitating sexual-health decision-making, access to vital prevention services such as pre-exposure prophylaxis (PrEP), as well as initiation of ART, improvement in HIV morbidity and mortality [11, 12, 19]. However, the stigma associated with HIV, sexual status, and gender expressions can combine with limited HIV knowledge and other factors to reduce interest in HIV testing among GBMSM. Nevertheless, there is limited research on the HIV prevention experiences of GBMSM or the social experiences that motivate or hamper HIV testing behaviors among GBMSM in Ghana. A better understanding of these factors may facilitate access to comprehensive HIV prevention services and ultimately

reduce disparities in HIV testing and incidence [2]. This qualitative study, therefore, seeks to understand the HIV testing experiences of Ghanaian GBMSM and factors that motivate or block willingness to test for HIV.

## Methodology

### Design

This study constitutes a part of a formative phase of a randomized controlled trial (RCT) that aims to assess the feasibility, acceptability, and potential effect size of a multi-level intervention to reduce intersectional stigma related to HIV, gender nonconformity, and sexual stigma and to increase HIV testing among GBMSM in Ghana [12]. Here, we provide a summary report of the study approach; a detailed description has been published elsewhere [10–12].

The assessment phase reported in this paper used a multi-method phenomenological design, which combined focus group discussions (FGD) and in-depth interviews (IDI) to understand experiences of HIV testing among GBMSM in Ghana. The included questions focused on HIV testing experiences, motivators and facilitators to HIV testing, and barriers to HIV testing at the intra and interpersonal levels, community levels, and healthcare facility levels. In addition to the FGD questions, the IDI also inquired about positive and negative experiences around HIV testing and diagnoses.

### Sampling and data collection

Two community partners in Accra and Kumasi employed a combination of convenience and snowball sampling to recruit GBMSM in Ghana. FGD participants were 18 years or older, male assigned at birth, self-identified as men, and had sex with other men within the previous six months. In addition, IDI participants also self-disclosed that they were living with HIV. The interviewers self-identified as gay and bisexual men from the respective cities and had received data collection and human subject research training. The partners invited GBMSM to participate in the IDIs (n10) and FGDs (n = 8) at their respective organizations. The IDIs and FGDs were primarily conducted in English, although two FGDs were conducted in Twi. The interviewers digitally recorded and transcribed all IDIs and FGDs. The research team reviewed both English and Twi audio files to ensure consistency. The research team reviewed the first several IDIs, provided the interviewers' feedback, and revised the interview guides as necessary.

**Ethical approval.** The institutional review boards of Yale University, Noguchi Medical Research Institute, the Ghana Health Service, and the University of Toronto approved the study. All study participants signed an informed consent form and received a token for participation. Only community-level partners had direct access to participants; they deidentified the data and shared it with the research team for coding and analysis. Hence, no information harmful to participants is accessible to any co-authors in this manuscript.

### Data analysis

Seven members from the research team and partner organizations coded the data line by line. First, all members reviewed the transcripts to identify relevant concepts that informed the construction of a codebook. Using the code book, two coders independently coded the same transcripts and compared their codes to reach a consensus regarding discrepancies in a meeting. Through an iterative process, two authors organized the codes by categories and subcategories based on the code book. We then combined all codes into broad classifications of participants' experiences and perspectives. We reviewed the full transcripts and

extracted quotes that fell under each category to report in the current paper. The research team examined the resulting groups of codes across all transcripts to confirm their comprehensiveness and representation.

## Results

We identified and analyzed the relationship and commonalities between the participants' responses to understand HIV testing experiences of GBMSM and factors that motivate or block willingness to test for HIV. Below, we explain the detailed results under major categories such as experiences with testing, motivations, and facilitators of testing, and barriers to testing.

### Experiences of HIV testing

**1) Fear of HIV infection created a stressful HIV testing experience.** Participants described their experiences during the HIV testing process and their feelings while getting tested. Some found the process quite "traumatic" and scary as they were unsure whether they would receive a reactive or nonreactive test result. As shown in participant accounts, these feelings of distress and fear are due to the automatic linkage of HIV to death and illness.

"I could remember when I went to do my HIV test; it was very scary; I was like, wow, ooh my gosh! What if I am HIV positive? What will happen next? So, when I did my test, the results didn't even come, but I was crying; I was crying, and I was so scared because I was like, Eish! (sound for fear or anxiety) so, if I am HIV positive, would I die? I would die because I can't take medicine for the rest of my life" (FGD)

"A whole lot was on my mind. Even within that five to ten minutes, I was sweating because I didn't know what the results would turn out to be. So, I was like, wow, oh God help me, don't let it be that I am HIV positive, so you know it was like a whole lot (overwhelming)." (FGD)

"Being HIV positive is not something you should be happy with, so when I was going for the test, I was having this panic attacks and anxiety" (IDI)

**2) A friendly and supportive HCF environment facilitated a positive HIV testing experience.** Some participants' experience during the HIV testing process was positive due to the HCF environment. The providers, such as nurses, were "friendly" and treated them equally." As seen in subsequent quotes, participants typically anticipate stigma due to negative reports from others on their experiences in HCFs. Hence, meeting friendly providers and feeling not discriminated against fostered a positive experience.

"There wasn't an obstacle because HIV test is free, the process of testing was quite easy, with nurses being friendly and treating us equally" (FGD)

"Because I was having these few friends, and they sometimes do meetings, and they sometimes bring nurses, and we talk about HIV and stuff, and they realize. . . they [Nurses] made us know that we are really at risk. So, it prompted me to give it a chance to do the test" (IDI)

### Motivators and facilitators of HIV testing

Participants who had tested for HIV discussed their reasons for getting an HIV test. Participants highlighted peer influence, their sexual orientation, sexual behaviors, receiving information about HIV, work requirements, and having symptoms as the significant reasons for testing.

**1) Perception of HIV testing as a prevention strategy served as a motivator for testing.** Some participants view HIV testing as a mechanism for preventing themselves and others from contracting HIV. Some indicated that, based on their lifestyle as men who have sex with other men, they were more at risk of contracting HIV. Hence, they needed to know their HIV status. Participants shared that testing provided peace of mind and helped protect themselves and others within their networks from being infected.

"Once you have accepted you are an MSM, you need to check your status. I'm saying this because if heterosexuals are getting tested to know their status, why not you, the MSM? I believe, as MSM, we should get tested." (FGD).

"So as an MSM person living, it's always good for you to at least go for your test every three months or every six months because you know that you are doing something that puts you at risk. So, your life is always at risk." (IDI)

"I think the very best thing that could happen to someone who is MSM is to test for HIV. Like in very simple and short words. That's what I tell people because once you know yourself and you know the risk you get exposed to once a while, or you know your identity, the best thing for you is to test for HIV." (FGD)

**2) Encouragement from friends and peers serves as a motivator for testing.** Some participants had overall positive experiences of peer support. Getting tested resulted from the encouragement and support from their peers. This support included providing information about HIV, sharing their own testing experiences, and accompanying them to get tested at the healthcare facility.

"A friend of mine who had gone in to test his HIV status encouraged me also to test and know my status." (FGD).

"I have friends who are peer educators, and they kept telling me that I should come so on the world AIDS day, two years ago I tested" (FGD)

"At first, I didn't want to test, but I changed my mind when I had education from a peer educator. That's how I had my first test." (FGD).

**3) Understanding the risk associated with sexual behaviors served as a motivator for testing.** Testing for HIV, according to some participants, was a critical need for GBMSM due to HIV risk behaviors. They identified behaviors such as concurrent partnerships, condomless sex, and possible partner unfaithfulness that may put them at risk, thus necessitating consistent HIV testing.

One participant shared his thoughts: "I don't have one sexual partner. I have multiple sexual partners, and as a guy or as an MSM having multiple sexual partners, I think it's better for me to test to know my status." (FGD).

"What helps me always to get tested is because I always know that I am an MSM, and sometimes I do love to have unprotected sex (sex without a condom), so I always like to go for testing at least every three months." (FGD)

"I have a partner, and I don't trust him for being faithful, so testing is important to me as an MSM (they discussed partners can be infected elsewhere)." (FGD)

**Transactional sex,** a source of income for some participants, was discussed as a reason to get tested. Participants who engaged in transactional sex thought it was essential to test for HIV. Implying that although they have sex for money and other benefits, they understand that it is vital to test for HIV as they do not intend to stop exchanging sex.

"I got tested because of my job (sex work). Yes, I see it as a job because I and some friends that I know depend on this to survive. Many of us don't work, so exchanging sex for money is what keeps us moving." (FGD).

"I know the kind of job I do (sex work), and I have no plans of stopping anytime soon. So, I think frequent testing is key" (FGD)

**4) Access to free testing and incentives facilitated testing.**   Monetary incentives added to free testing opportunities were a critical motivator for testing: As part of peer educator programs or research opportunities, community organizations often incentivize people who agree to participate. Hence, some went to test because of the incentives.

"I think it was a research program or something like that, and there was money involved. I think they were giving ₵30.00 or something, and I wanted to get the money, and I got there, and I was tested" (FGD).

Another participant shared, "I got tested because I wanted the 40 cedis. Trust me, it is the truth I am telling you" (FGD)

**5) Early symptoms and provider recommendations facilitated testing.**   Receiving a recommendation or instructions from a healthcare provider to test for HIV was critical to HIV testing decision-making. Per participant accounts, their providers recommend testing due to illness. Implying delay of testing until the point of illness. Some started experiencing symptoms such as fevers, general body weakness, weight loss, loss of appetite, headaches, rashes on the body, and discharges from the penis before they tested for HIV.

"I was not feeling well, and the doctor said I should go and do it (referral to HIV testing), so I just decided to do it." (FGD). "So there was a time the doctors were not figuring what was the sickness. So one of the doctors told me to go and run an HIV test. So I went [to get tested]." (FGD)

"Sometimes I take my bath, and my body itches me; That alone was enough for me, and I felt like I was also losing weight a bit. Though I am not the fat type, I felt I lost weight a little bit. Somebody would sometimes see you and act like, 'Are you sick?' and that kind of stuff so it alerted me and made me feel like, okay, let me go and test and see why I am having this kind of feeling. That is the reason why." (FGD).

"And you know, once in a while, you feel some rashes, and you feel a bit sick and that kind of stuff. Sometimes, you feel like you have lost your appetite and those kinds of things. So, I

really wanted to test so that I would really see what is really wrong with me." (IDI) "I had a discharge from my penis and headache" (IDI).

**6) Work or school requirements for testing.** Some participants shared that they got tested due to work policies or as an entry requirement for certain occupations such as the military and nursing school;

"Because of the nature of my work, I have to get tested. I am a caterer." (FGD).

"Well, it is the school's policy, so you have to do it whether you like it or not. Before you go to nursing school, you have to go for a medical report just to be fit for the training school. So, I got tested last two years (also a long time ago). I was like, this thing would come out, and what if I'm positive? Because I had never tested myself before, so I was thinking a lot, but it came out that I was not reacting. I was like, "Hurray!"" (FGD)

## Barriers to HIV testing

**1) Negative community perceptions of HIV deter testing.** Participants discussed how community understanding of HIV is critical to prevention needs. HIV as a disease was usually associated with immoral behaviors, as a punishment for deviant behaviors related to sex or contracted by only certain groups such as GBMSM or sex workers. Participants discussed that these prevalent community beliefs about HIV led to stigma, impacting their interest in HIV testing. When asked about factors that made it hard or difficult for them to engage with HIV testing, participants responded;

"They say it is [HIV] a satanic disease; you got this disease from fornication and whoring, and all sorts of stuff. On my first day, I told myself [NAME]! Be bold and go and test; if you are negative, then you are negative, and if you are positive, you know what to do." (FGD)

"In my community, they talk bad about HIV, especially when you are gay. They say it is MSM or gays who usually get infected with this kind of disease. So, they talk bad about that HIV." (FGD)

**2) Individual-level low-risk perception/indifference about HIV infection hinders testing.** In a world of competing health risks, some participants did not perceive HIV to carry a severe health risk. Some participants believed that many other diseases could kill them; thus, there was no need for the extra focus on HIV.

"Let me say, I am not interested in testing, or I have never tested or nothing because, as I said, I work in the health sector, and my philosophy in life is that all die be die cos working in the health sector and looking at the things that are killing people in Ghana, HIV is not top 10 killer disease" (FGD).

The most killer disease is RTA, Road Traffic Accidents, or malaria those days, so I don't see myself bothering about HIV or whatever. To me, I am in the hospital. I think if HIV will kill you, it will kill you. Even recently, corona [COVID-19] is in, and it is killing people, so what is the big deal? You understand it. HIV is still there, so I don't (test), I have never tested, and I don't think I will ever test." (FGD)

**3) Healthcare-level sexual stigma deters GBMSM from testing for HIV.** Some participants discussed the lack of MSM-friendly nurses and stigma-free providers as factors that

affect their interest in testing. Stigma from provider manifests through demeaning comments to GBMSM because of their engagement in same-gender partnerships and sexual activities. Thus, some avoid visiting some facilities for testing.

> "I think there was this one facility that my partner and I visited at that time. I heard some comments from one of the nurses that made me felt uncomfortable. So we refused to get tested at that time, so we had to leave. One of the nurses said, 'Your people are here' (referring to a more friendly colleague who typically helps GBMSM). So the friendly nurse responded, 'I know'. So, she added how can a nice guy like this be Gay?" (common questioning in Ghana to suggest wondering why someone is engaging in a behavior they consider wrong) (FGD).

> "The health provider told him (participant's friend) straight away, do you do men? All of a sudden, he was thrown aback. Why do you ask such a question? It is not only MSM who contract HIV. So, he tried to question the health provider, and the health provider said, 'Immediately I saw you, I knew you were one of them. They come here all the time, and we test them, I know how they behave' (FGD)

> "I know a lot of HIV testing sites that are not MSM-friendly because they were not having friendly Nurses. I had to move from my place to another place for the HIV test." (IDI)

**4) Other economic and health system issues.** The cost of transportation to the centers to get tested served as a barrier to getting tested. Participants discussed that since testing was free, the challenge usually involved paying for travel to the health centers. For some, despite their readiness to get tested, there was a lack of testing kits available at the facilities they visited over multiple months. When asked about any other barriers, participants mentioned the following:

> "I will say transportation will be the only cost, but there were no charges at the facility. And it was free. All I need to do is to be at the facility, and the test will be done. So, the only cost will be moving from my home to the facility to get tested." (FGD).

> "It was the shortage of test kits . . .I went to the facility, and I was told there wasn't any test kit." (FGD).

## Discussion

Limited evidence from Ghana and SSA shows that despite consistently a disproportionately high burden of HIV, GBMSM infrequently test for HIV and often delay testing until the point of illness [22–26]. This study, therefore, provides qualitative insights into experiences, motivators, and barriers to HIV testing among GBMSM. The findings point to important issues around stigma, education, peer support, and healthcare resources that affect HIV testing and can inform interventions to increase testing among GBMSM in the country.

We established that stigma associated with HIV and same-sex intercourse increases negative experiences with testing and dissuades GBMSM from testing. Consistent with previous studies, participants' accounts show how stigma at the healthcare-, community-, and individual levels affects testing experiences and discourages GBMSM from testing [20, 27]. At the community level, GBMSM recount the association of HIV with sin, same-sex intercourse, and sex work; at the healthcare level, they experience negative comments and intrusive questioning around sexual behavior. The accounts by participants corroborate the limited findings in Ghana and West Africa that show HIV and same-sex intercourse stigma as factors that affect

HIV testing and use of HIV services among GBMSM in the region [10, 23, 28]. As explained by previous findings, participants in this study internalized HIV stigma. They began to develop a fear of HIV, thus experiencing stress and fear during HIV testing and or avoiding testing in some facilities [10, 23, 28].

Despite the adverse effects of stigma, we found that a positive healthcare environment can provide better experiences and encourage HIV testing among GBMSM. Participants who reported positive experiences with testing mentioned the professionalism and confidentiality of providers at the testing sites as factors that contributed to their positive experience. Likewise, those who avoid testing sites mentioned that stigma from providers prevents them from testing. Our findings on a positive healthcare environment as a facilitator for testing confirm findings among GBMSM elsewhere [29–31]. It also explains why GBMSM from previous studies reported a preference for friendly and GBMSM-led healthcare facilities for HIV services, compared to government facilities where stigma appears more rampant [31, 32]. Therefore, we reaffirm the importance of implementing interventions to alter the stigma and negative experiences of GBMSM in public healthcare facilities to increase HIV testing and care [12, 13, 23].

Our findings highlight the importance of increasing education and proper information channeling to GBMSM in Ghana to increase HIV testing. The participants who expressed an understanding of HIV risk associated with same-sex intercourse, HIV transmission, and the role of HIV testing mentioned those as factors that informed their decision to test for HIV. Conversely, participants who expressed a lack of awareness, individual-level low-risk perception, and indifference about HIV infection did not test for HIV. Whereas limited interventions or studies focused on GBMSM in Ghana and sub-Saharan African countries, we previously implemented an adapted Many Men Many Voices intervention in Ghana that addressed knowledge and capacity around HIV risk, transmission, prevention, and care among GBMSM [13, 15, 33]. The findings of this study support our initial evidence from the Many Men Many Voices intervention that showed that HIV testing and readiness to test for HIV increased after receiving education and information about HIV [13, 15, 33]. Similar findings appeared in other studies in different parts of West Africa and SSA regions [34, 35].

Moreover, per our findings, peers can serve as promoters of HIV testing among GBMSM. As evidenced in this study, some participants tested for HIV because they were referred to testing by their friends and acquaintances and peer educators who contacted GBMSM to encourage testing. These findings confirm previous findings where a peer support intervention implemented among GBMSM addressed individual-level challenges and increased HIV testing [35–37].

Additionally, despite recent efforts through several programs (by Researchers, PEPFAR, and Ghana Health Services) to increase HIV testing among GBMSM, access remains affected by the cost and location of facilities for some GBMSM [3, 4]. Some participants indicated they could not afford to travel to the location where they could access HIV testing; others said they could not pay for HIV testing nearby. Therefore, emphasizing the urgent need to continue to increase support for programs that provide targeted HIV testing services to GBMSM free of charge. The issue of location and traveling can also be remedied by using newly available HIV self-testing recently introduced in Ghana [15, 38]. HIV self-testing, if implemented through existing peer networks, can provide easy access to GBMSM irrespective of their locations [15].

The need for interventions that address stigma and HIV testing needs among GBMSM is extremely important because of the healthcare facility level stigma and individual level factors and the national outlook around same-sex intercourse [10, 23]. Unlike their counterparts elsewhere, currently, GBMSM face various physical harm and criminalization by law in Ghana, which further pushes them to remain hidden [39–41]. In our work around the victimization and criminalization of GBMSM in Ghana, we found that a current anti-LGBTQI+ bill in

Ghana that intends to harden punishment for LGBTQI+ persons negatively affects GBMSM in the country [39, 40]. The bill has increased stigma and violence towards GBMSM and thus could negatively impact interest in HIV testing and care services [39, 40].

We implore researchers to consider the following limitations in interpreting our findings. As typical of qualitative studies and nonprobability sampling, the convenience and snowball sampling in the data collection limits its generalizability for GBMSM in Ghana [41, 42]. We also recruited GBMSM specific to Accra and Kumasi, Ghana's two largest urban settings; thus, findings may not represent other localities in Ghana. Additionally, this study was conducted to inform the development of an intervention to address intersectional stigma and increase HIV testing, thus limiting the scope to testing and no other aspects of care [12]. Hence, the findings should be used in addition to other findings on HIV testing and care among GBMSM in the geographical region. Nonetheless, the study provides an updated important insight into barriers, motivators, and facilitators of HIV testing, which, when considered, will help in understanding ways to increase access to HIV testing among GBMSM in Ghana and ultimately help Ghana reach its 95 95 95 HIV targets.

## Author Contributions

**Conceptualization:** Gamji Rabiu Abu-Ba'are, Gloria Aidoo-Frimpong, Laura Nyblade, Kwasi Torpey, LaRon E. Nelson.

**Data curation:** Gamji Rabiu Abu-Ba'are, Melissa Stockton, Richard Panix Amoh-Otoo, LaRon E. Nelson.

**Formal analysis:** Gamji Rabiu Abu-Ba'are, Gloria Aidoo-Frimpong, Edem Yaw Zigah, Samuel Amuah, Prince Amu-Adu, Richard Panix Amoh-Otoo.

**Funding acquisition:** Laura Nyblade, Kwasi Torpey, LaRon E. Nelson.

**Investigation:** Gamji Rabiu Abu-Ba'are, Edem Yaw Zigah, Samuel Amuah, Prince Amu-Adu, Richard Panix Amoh-Otoo, Laura Nyblade, Kwasi Torpey, LaRon E. Nelson.

**Methodology:** Gamji Rabiu Abu-Ba'are, Laura Nyblade, Kwasi Torpey, LaRon E. Nelson.

**Project administration:** Gamji Rabiu Abu-Ba'are, Prince Amu-Adu, Richard Panix Amoh-Otoo, Laura Nyblade, Kwasi Torpey, LaRon E. Nelson.

**Supervision:** Gamji Rabiu Abu-Ba'are, Laura Nyblade, Kwasi Torpey, LaRon E. Nelson.

**Validation:** Gamji Rabiu Abu-Ba'are, Melissa Stockton, Laura Nyblade, Kwasi Torpey, LaRon E. Nelson.

**Writing – original draft:** Gamji Rabiu Abu-Ba'are, Gloria Aidoo-Frimpong, Melissa Stockton, Edem Yaw Zigah.

**Writing – review & editing:** Gamji Rabiu Abu-Ba'are, Gloria Aidoo-Frimpong, Melissa Stockton, Edem Yaw Zigah, Samuel Amuah, Prince Amu-Adu, Richard Panix Amoh-Otoo, Laura Nyblade, Kwasi Torpey, LaRon E. Nelson.

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
