## [Decision Letter · Decision Letter 0]

10 Oct 2023

PGPH-D-23-00411

“I told myself, be bold and go and test”: Motivators and barriers to HIV testing among gays, bisexuals, and all other men who sex with men in Ghana – West Africa

Dear Dr. Abu-Ba'are,

Thank you for submitting your manuscript to PLOS Global Public Health. After careful consideration, we feel that it has merit but does not fully meet PLOS Global Public Health’s publication criteria as it currently stands. Therefore, we invite you to submit a revised version of the manuscript that addresses the points raised during the review process.

We look forward to receiving your revised manuscript.

Kind regards,

Kevin Escandón, MD, MSc

Academic Editor

Journal Requirements:

2. In the online submission form, you indicated that "The datasets used and/or analyzed during the current study are not publicly available due to our ethical and legal requirements related to protecting participant privacy and current ethical institutional approvals but are available from the corresponding author on reasonable request pending ethical approval". All PLOS journals now require all data underlying the findings described in their manuscript to be freely available to other researchers, either 1. In a public repository, 2. Within the manuscript itself, or 3. Uploaded as supplementary information.

Editor Comments:

This manuscript has scientific value but needs major rewriting in several sections and addition of methodology details. Please address all comment one by one in your itemized response letter.

Reviewers' comments:

Reviewer's Responses to Questions

**Comments to the Author**

1. Does this manuscript meet PLOS Global Public Health’s publication criteria? Is the manuscript technically sound, and do the data support the conclusions? The manuscript must describe methodologically and ethically rigorous research with conclusions that are appropriately drawn based on the data presented.

Reviewer #1: Yes

Reviewer #2: Yes

2. Has the statistical analysis been performed appropriately and rigorously?

Reviewer #1: N/A

Reviewer #2: N/A

3. Have the authors made all data underlying the findings in their manuscript fully available (please refer to the Data Availability Statement at the start of the manuscript PDF file)?

Reviewer #1: No

Reviewer #2: No

4. Is the manuscript presented in an intelligible fashion and written in standard English?

Reviewer #1: Yes

Reviewer #2: Yes

5. Review Comments to the Author

Reviewer #1: Overall

Clear and straightforward results that make a contribute to the literature on what can facilitate or prevent HIV testing among men who have sex with men in Ghana (i.e., a priority population in a less-studied part of the world).

If you want to cite a prior paper (authored by me!) that uses similar qualitative methods to reach quite similar findings among men who have sex with men in New York City, please see Kobrak, et al., 2022: https://pubmed.ncbi.nlm.nih.gov/35536518/

Abstract, beginning of Results, and the Discussion repeat a summary of the same findings.

“There is limited research on the <hiv experiences="" prevention=""> of GBMSM or into the factors that motivate or hamper HIV testing behaviors among GBMSM in Ghana.”

- Are you only looking at men’s HIV prevention experiences? How about those sexual or social experiences that relate to stigma and can affect testing?

Perhaps specify what the token was for participation.

Some language

- “Gays” and “bisexuals” sound odd in U.S. English

- In U.S., standard term is “assigned male sex at birth”

- “To assess the feasibility, acceptability, and potential effect size of a multi-level intervention to reduce intersectional HIV, gender nonconformity, and sexual stigma and to increase HIV testing among GBMSM in Ghana”

o Maybe instead, “Stigma related to… intersectional HIV”

- “GBMSM disproportionately account for new HIV infections” seems inexact and maybe even a bit stigmatizing; maybe ““disproportionately acquire HIV” or “experience a disproportionate burden of HIV.”

- Clarify if IDIs are with individual men

- Could make content and language more concise and avoid imprecise passive voice constructions: “While HIV testing practices are slowly increasing among GBMSM in Africa,[2, 19] there is a need to understand the factors the affect HIV testing behaviors for this vulnerable group, and to begin to address such factors” � “Further increasing HIV testing among GBMSM in Africa[2, 19] requires that we understand and address factors that can affect men’s HIV testing practices.”

- This method and framework needs explanation, and is unclear without reading the citations: “Recognizing the centrality of stigma in HIV testing practices, this study employs the Health Stigma and Discrimination Framework (HSDF) to unpack GBMSM’s experiences across the socio-ecological model.[20] HSDF posits that the stigmatization process unfolds across the socio-ecological spectrum in the context of a health condition, such as HIV care, and ultimately impacting uptake of testing, adherence and treatment, incidence, morbidity and mortality among affected groups [20].”

Abstract

“infrequently often”?

Maybe say “perceived stigma at Healthcare facilities”

Define acronyms at first use

Introduction

Lots of data from another study: “For instance, condom use was higher when GBMSM had vaginal sex with women, (rel. f. = 0.75) than anal sex with men (rel. f. = 0.50) [14]. About 40% of did not use condoms during first sexual encounters, and even less (38%) continuously used condoms with sexual partners [14]. These increased risk behaviors and high HIV prevalence among GBMSM shows the need for increased HIV testing [10-16].”

Aims: Focus on stigma?

Method / Design / Data Analysis

You describe coding in detail. To complete the methodological description, could you add how you selected the coded content to include in the article?

Results

Some of the barriers and facilitators very succinctly described in quotes could perhaps benefit from longer quotes or additional narrative that more fully describe the participant’s thinking and situation.

- For example, here is a great, counter-narrative quote that is quite long and gives the reader a sense of the participant’s thinking: "Let me say, I am not interested in testing or I have never tested or nothing because as I said I work in the health sector and my philosophy in life is that, all die be die cos working in the health sector and looking at the things that are killing people in Ghana, HIV is not top 10 killer disease… I have never tested and I don't think I will ever test.”

I wondered if the quotes were directly from transcripts of interviews and focus groups. For example, do men in Ghana routinely refer to themselves as “MSM” as in quotes? e.g., “Once you have accepted you are an men who have sex with men…”; “As a guy or as an GBMSM having multiple sexual partners…”

The introduction refers to “As such, there is lower use of prevention tools such as condoms, lubricants and HIV testing, and in some cases, limited adherence to medication.”

- In the results, I saw limited results about the specific ways men have sex with other men that may put them at risk of HIV acquisition or influence their approach to HIV testing – e.g., few mentions of whether men use condoms or PrEP.

- Are there other findings on this that it would make sense to add?

- Is PrEP widely available to men in Ghana?

Maybe explain a bit the importance of "treating them equally” i.e., did participants find that a lack of negativity or discrimination towards gay or femme-presenting men give them positive feelings that made testing easier?

This seems like a very important quote about how to understand or encourage HIV testing, but I think it needs explanation: “I think the very best thing that could happen to someone who is MSM is to test for HIV like in very simple and short words. “

“HIV as a disease was usually historically associated with immoral behaviors, as a punishment for deviant behaviors related to sex or contracted by only certain groups such as MSM or commercial sex workers.”

The effect of peer educators and testing providers promoting positive messaging about HIV testing seems so important. How did this work come about? Who built these programs?

Organization: Perhaps, given the overlap between Results sections

- Heading 2: Could call it “Other Motivators and Facilitators of HIV testing”

- Heading 3: Understanding risk associated with sexual behaviors such as transactional sex served as a motivator for testing. But what followed did not describe experiences of transactional sex

Maybe indent quotes to avoid having to mark quotes inside quotes

Discussion

For the the Discussion could expand on the National HIV context: I was wondering, do men understand HIV as a disease that can be effectively treated? Has the situation changed in Ghana in recent years? Is HIV treatment now more widely available? Do men see their peers on HIV treatment thriving? (I see in Discussion: “thus limiting the scope to testing and no other aspects of care”)

LGBTQ acceptance context: Any sense of whether there is growing acceptance among the public and the professional medical class in Ghana towards gay and femme men?

This statement doesn’t seem to match the experience of many of the men motivated to test: “As explained by the HSDF and previous findings, participants in this study internalized the stigma and begin to develop fear for HIV, thus experiencing stress during HIV testing, and or avoid testing in its totality [10, 23, 28].”</hiv>

Reviewer #2: The investigators report the results of a qualitative study the barriers and facilitators to HIV testing among Gay, Bisexual and men who has sex with men in Ghana. The study is positioned to inform the design of a future randomized pilot study. The paper is well-written with a clear message. I have several comments that would improve the quality of the manuscript.

Abstract:

Several abbreviations are used but not spelled at. Please write the following in full at first use: SSA, GBMSM, IDI, FGD.

The sampling approach is described, but there is no mention of the study design.

Stating that something happens infrequently often is contradictory and confusing.

Main text:

Please report all abbreviations at first use.

It is unclear what “rel.f.” refers to. See line 64.

How was the sample size determined? Did you stop at data saturation?

Why were both indepth interviews and focus groups needed?

It would be helpful the highlight the themes identified by underlining them or using italics as allowed by the journal.

Consider putting each quote on a new line. This improved readability.

Overall, this is a useful contribution to the literature.

6. PLOS authors have the option to publish the peer review history of their article (what does this mean?). If published, this will include your full peer review and any attached files.

**Do you want your identity to be public for this peer review?** For information about this choice, including consent withdrawal, please see our Privacy Policy.

Reviewer #1: No

Reviewer #2: No

---

## [Decision Letter · Decision Letter 1]

28 Nov 2023

“I told myself, be bold and go and test”: Motivators and barriers to HIV testing among gay, bisexual, and cis-gender men who sex with men in Ghana – West Africa

PGPH-D-23-00411R1

Dear Dr. Abu-Ba'are,

We are pleased to inform you that your manuscript '“I told myself, be bold and go and test”: Motivators and barriers to HIV testing among gay, bisexual, and cis-gender men who sex with men in Ghana – West Africa' has been provisionally accepted for publication in PLOS Global Public Health.

Best regards,

Kevin Escandón, MD, MSc

Academic Editor

Reviewer Comments (if any, and for reference):

Reviewer's Responses to Questions

**Comments to the Author**

1. If the authors have adequately addressed your comments raised in a previous round of review and you feel that this manuscript is now acceptable for publication, you may indicate that here to bypass the “Comments to the Author” section, enter your conflict of interest statement in the “Confidential to Editor” section, and submit your "Accept" recommendation.

Reviewer #2: All comments have been addressed

2. Does this manuscript meet PLOS Global Public Health’s publication criteria? Is the manuscript technically sound, and do the data support the conclusions? The manuscript must describe methodologically and ethically rigorous research with conclusions that are appropriately drawn based on the data presented.

Reviewer #2: Yes

3. Has the statistical analysis been performed appropriately and rigorously?

Reviewer #2: N/A

4. Have the authors made all data underlying the findings in their manuscript fully available (please refer to the Data Availability Statement at the start of the manuscript PDF file)?

Reviewer #2: No

5. Is the manuscript presented in an intelligible fashion and written in standard English?

Reviewer #2: Yes

6. Review Comments to the Author

Reviewer #2: Thank you for addressing my comments.

7. PLOS authors have the option to publish the peer review history of their article (what does this mean?). If published, this will include your full peer review and any attached files.

**Do you want your identity to be public for this peer review?** For information about this choice, including consent withdrawal, please see our Privacy Policy.

Reviewer #2: No
